# On the Road Safety: Gender Differences in Risk-Taking Driving Behaviors Among Seniors Aged 65 and Older

**DOI:** 10.3390/geriatrics9050136

**Published:** 2024-10-21

**Authors:** Pierluigi Cordellieri, Laura Piccardi, Marco Giancola, Anna Maria Giannini, Raffaella Nori

**Affiliations:** 1Department of Psychology, Sapienza University of Rome, 00185 Rome, Italy; pierluigi.cordellieri@uniroma1.it (P.C.); annamaria.giannini@uniroma1.it (A.M.G.); 2San Raffaele Cassino Hospital, 03043 Cassino, Italy; 3Department of Biotechnological and Applied Clinical Sciences, University of L’Aquila, 67100 L’Aquila, Italy; marco.giancola@univaq.it; 4Department of Psychology, University of Bologna, 40127 Bologna, Italy

**Keywords:** risk-taking driving, older drivers, pedestrians, car accidents, sex differences

## Abstract

**Background/Objectives**: Life expectancies have increased in most countries, leading to a higher accident rate among older drivers than their younger counterparts. While numerous studies have analyzed the decline in cognitive abilities and physical limitations as contributing factors, there are other considerations. For instance, younger male drivers tend to take more risks than younger female drivers. However, there is a lack of research and evidence regarding the role of gender in risk-taking among individuals over 65. Given this gap, our current study aims to investigate the relationship between gender and risk propensity in this particular age group. The primary goal was to determine if driving experience affects the gender gap in risk attitude; **Methods**: We studied risk behavior in both car drivers and pedestrians. Our sample included 200 individuals (101 women), all over 65, with the same weekly driving times. After a brief demographic and anamnestic interview, they completed the Driver Road Risk Perception Scale (DRPS) and the Pedestrian Behavior Appropriateness Perception Scale (PBAS) questionnaires. They also provided information about traffic violations and road crashes; **Results**: Our research revealed that older male drivers continue to tend to risky behavior, highlighting the need for targeted interventions to improve risk awareness, especially among older men; **Conclusions**: Our findings suggest that road safety messages should specifically target male drivers as they are less likely to view responsible driving actions, such as observing speed limits, as desirable.

## 1. Introduction

Road accidents have been identified as one of the primary contributors to fatalities. Research has extensively reported that various factors can affect driving behavior, such as age, gender, experience, and both physiological and psychological features. However, age and gender stand out as the most measurable indicators. In 2018, the World Health Organization (WHO) reported that, worldwide, 29% of road fatalities involved car occupants, with males being three times more likely than females to suffer fatal injuries at all ages [1].

The gender gap in serious road accidents is still a problem, even when considering factors like mileage and road type [2]. Numerous studies have consistently indicated that males commit more traffic violations and take more significant road risks than females [3]. This result holds especially true for speeding, as evidenced by research conducted by Cestac et al. [4]. One of the most noticeable gender disparities between males and females regarding driving behavior is that men tend to exhibit more anger and risk-taking. In contrast, women tend to display more anxiety and distraction while driving [5].

Furthermore, males are overrepresented in crash rates due to decreased compliance with traffic rules [6,7]. Gonzalez-Iglesias et al. [8] found that men receive more fines and violate traffic rules more often than women, despite some evidence suggesting that women might commit specific violations more frequently than men [9]. Furthermore, the over-representation of males in car crashes may be attributed to females driving less than males [10].

The higher incidence of male involvement in road accidents can be attributed to biological and psychosocial factors. Men often struggle to control impulsive behavior due to lower cognitive control. They also tend to conform to gender roles and stereotypes related to capability and risk-taking. Numerous studies have revealed the presence of distinct gender stereotypes that portray males as more adept at navigating cars, even when exhibiting riskier driving behavior and more transgressions compared to females [11,12,13]. These beliefs often justify men’s tendency to engage in risky driving behavior, citing their superior driving skills.

On the other hand, risk-taking driving behaviors and traffic rule violations perpetrated by females are interpreted as a lack of capability in driving. The root cause of road accidents lies in aggressive driving behavior. In contrast, other behaviors, such as anxiety or excessive caution, disrupt the regular flow of traffic, resulting in congestion and reduced efficiency. From this perspective, both types of drivers pose serious problems that impact road safety.

However, overlooking other factors that could add context, influence, or alter the risk trend would be a mistake. Although experience directly affects driving ability, one cannot discount the impact of a driver’s age on their likelihood of taking risks. Various studies have indicated that young males, in particular, tend to display a higher propensity for risk-taking behaviors [14,15].

Several studies have repeatedly demonstrated that individuals under 25 are more inclined to partake in hazardous driving practices than their older counterparts [16]. While it is widely recognized that turning 25 marks a significant neurodevelopmental milestone for the prefrontal cortex [17], it is crucial to acknowledge that not all tendencies toward risk-taking dissipate at this age. Unfortunately, there exists a need for more research investigating the propensities of individuals in their late twenties and early thirties to engage in reckless driving behaviors, as this represents a critical age group to study. McCartt et al. [18] found that older and younger drivers have different crash rates, with younger drivers reporting more crashes. Jimenez-Mejıas et al. [19] observed 1574 students for three consecutive academic years to evaluate patterns in risky driving behaviors. Their longitudinal results confirmed that male drivers often participated in dangerous driving behaviors. Curry et al. [20] highlighted how young drivers of the same age but differing driving experiences exhibit differing crash rates. Their findings demonstrated that 21-year-old novice drivers were involved in more road crashes than experienced drivers of the same age. At the same time, both groups reported notably fewer collisions compared to novice drivers aged 17 to 20. In general, drivers under 30 are more willing to overtake cyclists and buses and go through an amber traffic light than drivers over 30. The younger group consistently exhibited a greater inclination to accelerate when confronted with risky situations than the older group. Extensive research has delved into the connection between drivers under the age of 25 and their propensity for engaging in risky driving behaviors, primarily due to the over-representation of this age cohort in road crashes [21,22].

Furthermore, research indicates that males exhibit higher levels of sensation seeking, characterized by a propensity to pursue intense, novel, or potentially risky activities. Conversely, females have been found to have a lower tolerance for delayed rewards. These findings are supported by various studies [23,24,25,26].

Few studies have shown how age and driving experience affect the likelihood of engaging in risky situations. It is well known that young male drivers tend to be more optimistic about their driving ability than older male drivers and are more willing to engage in risky behavior [27]. Moreover, they tend to underestimate the risks associated with particular traffic situations, such as speeding or using a phone while driving, compared to older drivers [28]. In light of this, there may be a case for a differential impact of age and driving experience on the inclination to partake in risky driving behavior.

Chen et al. [29] compared two driver groups: one composed of individuals aged 18–60 years and the other consisting of individuals over 60. The study revealed that older adults’ reaction time was significantly higher than younger groups. Additionally, the researchers observed that older drivers with more experience exhibited decreased lateral stability, particularly when compared to their younger counterparts. This increased variability in lateral deviation made them more prone to colliding with obstacles on the right side of the road. Interestingly, older drivers with more experience demonstrated better performance in terms of speed deviation than their younger counterparts. Despite this, the study concluded that older drivers exhibited less lateral stability than younger groups, albeit in opposite directions. Consequently, these findings illustrate that ample driving experience primarily aids in controlling driving speed, as opposed to maintaining sideways vehicle stability.

In the Italian context, older drivers are significantly more susceptible to fatalities compared to younger drivers. For instance, in 2016, drivers aged 18–24 were involved in 352 fatalities (rate of 3370 across Europe), while drivers over 65 had 1045 fatalities (rate of 7071 in Europe; European Transport Safety Council, 2019). These data indicate an increased risk propensity and potential impairments in sensory, cognitive, and physical abilities [30]. Interestingly, most studies on driving by the elderly have focused on analyzing the impact of cognitive impairment on driving safety, while few have examined changes in risk tolerance among individuals over 65. The growing population of active elderly people who rely on driving as a means of transportation demands further attention. Many in this age group remain employed, while others play a significant social role, often accompanying their grandchildren to various activities. Furthermore, the risk extends beyond just driving and encompasses pedestrian behavior. According to a study by Liu et al. [31], middle-aged pedestrians in Hong Kong faced the gravest danger of fatal or severe injuries in the initial year of the COVID-19 lockdown in 2020.

Researchers widely agree that the severity of pedestrian injuries varies by age, with elderly pedestrians being at a greater risk of sustaining severe or fatal injuries. This increased vulnerability is primarily due to age-related declines in cognitive function and physical fitness, which impair their ability to react to fast-moving vehicles and avoid potential collisions [32,33,34,35]; therefore, the accident statistics related to the elderly call for a comprehensive investigation. One crucial aspect to consider is whether the difference in risk tolerance between males and females decreases or persists.

The primary aim of this study was to analyze the gender gap in risk-taking behavior among a sample of 200 individuals aged 65 and above. Additionally, we thoroughly investigated the correlation between participants’ involvement in accidents and their risk tendencies while driving.

## 2. Materials and Methods

### 2.1. Participants

This survey was conducted with an opportunistic sample of 204 older individuals (age M = 70.59 years; SD = 4.36 years; education M = 11.49 years; SD = 3.98 years; 102 females). All participants were members of Italian local over-65 associations and declared that they had a driving license. Both groups reported identical weekly driving times and the same use of the car (see Table 3). They were all still actively involved in life, with no reported neurological or psychiatric disorders during an initial brief anamnestic interview. In addition, all the participants stated that they did not take drugs that impair driving for the treatment of any illnesses. The *z*-test, with ±4 *z* scores as the reference values for samples > 100 [36,37], indicated 4 outliers, which were discarded from the dataset. The final sample consisted of 200 participants (Mean_age_ = 70.54 years; SD_age_ = 4.37 years; Mean_education_ = 11.51 years; SD_education_ = 3.97 years; 101 females). Table 1 reports the main features of the research sample by gender.

### 2.2. Measures

The measures employed encompassed a variety of distinct variables, such as driving expertise (years of driving license, weekly car use), driving risk behaviors (Driver Road Risk Perception Scale, DRPS), traffic violation variables (driving accidents, responsibility in accidents, contributory negligence), and pedestrian risk behavior (Pedestrian Behavior Appropriateness Perception Scale, PBAPS).

#### 2.2.1. Driving Risk Behaviors

Driver Road Risk Perception Scale (DRPS). DRPS is a 13-item scale (see Appendix A) developed to measure the perception of driving risk through a wide array of driving conditions, such as mobile phone use (e.g., “typing a message on your mobile phone while driving”), eating (e.g., “eating in the car: sandwich, snacks, etc.”; “driving under conditions of significant physical fatigue”), and other specific risk behaviors (e.g., “Arguing animatedly with a passenger while driving”). Responses are rated on a 5-point Likert scale ranging from “not at all risky” (1) to “extremely risky” (5). The items used in this scale were derived from previous works by Fraschetti et al. and Cordellieri et al. [38,39]. The structure of the DPRS was tested by exploratory factor analysis (EFA) and confirmatory factor analysis (CFA). In EFA, the principal component analysis method was used to evaluate factor extraction, while the varimax method was performed for rotation. The Keiser–Meyer–Olkin (KMO) coefficient was 0.95, while the Chi-square value for Bartlett’s Sphericity Test was 3960.85 (*p* < 0.001). All extraction values were above the threshold of 0.30, and all diagonal values in the anti-image correlation matrix were above 0.50. A unidimensional structure was revealed, explaining 78% of the variance. As for CFA, because the data did not meet the assumption of multiple normality (thresholds for skewness and kurtosis: <2| and <7|, respectively; [40,41]), bootstrap maximum likelihood (ML) with 800 resamples was performed. The model fit indices were as follows: minimum discrepancy function by degrees of freedom divided (CMIN/DF) (122.722/54) = 2.273, goodness of fit Index (GFI) = 0.921, incremental fit index (IFI) = 0.982, Tucker–Lewis Index (TLI) = 0.974, comparative fit index (CFI) = 0.982, root mean square residual (RMSEA) = 0.078. The model fit indices ranged from acceptable to good limits. All items of the DRPS showed statistically significant path coefficients (*p* < 0.001). The internal consistency reliability was excellent: Cronbach’s α (0.97) and McDonald’s ω (0.97). Additionally, all items in the DRPS displayed excellent item-total correlation coefficients, exceeding the threshold of 0.30.

#### 2.2.2. Traffic Violation Variables

Three items were used to inquire about this: (1) driving accidents “In how many accidents have you been involved?”; (2) responsibility in accidents “How many times have you been responsible for a traffic accident?”; and (3) contributory negligence “In how many accidents were you found to have contributory negligence?”.

#### 2.2.3. Pedestrian Risk Behavior

Pedestrian Behavior Appropriateness Perception Scale (PBAPS). This uses a 7-item scale (see Appendix A) developed to assess the perceived appropriateness of various behaviors of elderly pedestrians in the road environment. The scale includes 7 items, each of which evaluates a specific behavior, such as being distracted on a cell phone (e.g., reading a text message on the phone while crossing the street if the street has little traffic) or engaging in risky conduct (e.g., crossing when the pedestrian light is red, if you are in a hurry). Respondents rate the appropriateness of each behavior on a 10-point Likert scale ranging from “not at all appropriate” (1) to “completely appropriate” (10).

The structure of the PBAPS was tested by EFA and CFA factor analyses. In EFA, the KMO coefficient was 0.90, while the Chi-square value for Bartlett’s Sphericity Test was 1068.52 (*p* < 0.001). All extraction values were above the threshold of 0.30, and all diagonal values in the anti-image correlation matrix were above 0.50. A unidimensional structure was revealed, explaining 69% of the variance. As data did not meet the assumption of multiple normality, CFA was computed using bootstrap ML with 800 resamples. The model fit indices ranged from acceptable to good limits, as follows: CMIN/DF (23.705/10) = 2.371, GFI = 0.970, IFI = 0.987, TLI = 0.972, CFI = 0.987, RMSEA = 0.080. Path coefficients for all items were statistically significant (*p* < 0.001), and Cronbach’s *α* (0.92) and McDonald’s *ω* (0.92) revealed excellent internal consistency reliability. In addition, all items in the PBAPS revealed excellent item–total correlation coefficients (>0.30).

## 3. Results

### 3.1. Data Screening and Preliminary Analysis

Statistical analyses were performed using SPSS Statistics version 24 for Windows (IBM Corporation, Armonk, New York, NY, USA). Shapiro–Wilk’s test indicated that all continuous variables were not normally distributed. Due to the non-normality of the data, Spearman’s correlation analysis was undertaken to ascertain the interrelationships among all the study variables. Table 2 reports all correlations among the study variables by gender.

### 3.2. Gender Differences in Expertise and Risk Behaviors

Furthermore, we employed odds ratios (ORs) to examine gender differences across various variables, including years of holding a driving license, weekly car use, risky driving behavior, driving accidents, responsibility in accidents, contributory negligence, and risky pedestrian behavior. The odds ratio is a statistical measure that quantifies the strength of the association between two events and is frequently utilized in survey research, epidemiology, and clinical studies, particularly in case–control designs.

In this study, our primary objective in using odds ratios was to evaluate the differences in risk propensity between males and females. To this end, we selected a threshold of one *z*-score above the mean (1 *z*-score = 84.1% of the sample). We posited that a higher score at this threshold, within variables related to road risk, signifies a high-risk condition, whereas a lower score indicates a low-risk condition. Accordingly, we categorized responses as high-risk or low-risk within both the female and male populations.

The odds ratio revealed a significant difference between male and female participants for the years of holding a driving license (OR = 2.83, 95% CI [0.971–8.27]), while no significant differences were found for weekly car use (OR = 0.660, 95% CI [0.108–4.04]). Regarding the risk-taking variables, the odds ratios indicated a statistically significant difference in the likelihood of engaging in risky behavior between females and males for driving risk behaviors (OR = 5.25, 95% CI [1.71–16.1]), driving accidents (OR = 5.03, 95% CI [ 1.82–13.9]), responsibility in accidents (OR = 2.71, 95% CI [1.07–6.86]), contributory negligence (OR = 2.26, 95% CI [1.11–4.63]), and pedestrian risk behaviors (OR = 3.34, 95% CI [1.17–9.58]). Table 3 shows that males exhibited higher years of holding a driving license, driving risk behaviors, driving accidents, responsibility in accidents, contributory negligence, and pedestrian risk behaviors than females. Figure 1 summarizes graphically the results of the odds ratio analysis.

## 4. Discussion

Taking into consideration the substantial increase in human life expectancy and the evidence that the European Union (EU) is renowned as the region with the world’s oldest population [42], with a staggering aging rate of 94.1% in 2001, which further surged to 125.8% in 2017 [43,44], it appears increasingly important to pay attention to the elderly population. European countries are experiencing population aging due to increased life expectancy and declining birth rates. Recent data from Eurostat (https://ec.europa.eu/eurostat/web/products-eurostat-news/w/DDN-20240503-2, accessed on 11 July 2024) indicates that Italy’s life expectancy exceeds the EU average following the pandemic.

Undoubtedly, the growing number of elderly drivers and their extended life expectancy have raised considerable concerns about road safety. Research has shown that older individuals are more likely to be involved in severe accidents compared to younger drivers, mainly because of their increased physical vulnerability and declining cognitive and sensory abilities associated with aging.

Our study contributes to the overall effort of accident prevention and safeguarding vulnerable groups. In particular, we focused on studying the risk-taking behavior of older drivers, while considering gender differences. We aimed to determine if the patterns observed in young male drivers persist in elderly individuals. In the Italian context, it is evident that older drivers face a significantly higher risk of fatalities compared to their younger counterparts.

Our data demonstrate that the gender disparity in driving risk-taking remains consistent even among individuals over 65. Men are more inclined to take risks than women, as indicated by their involvement in more accidents. This evidence highlights their perception of risk and reveals a distinct pattern of risk-taking driving behavior. These findings indicate that males exhibit a higher propensity for risk-taking as both drivers and pedestrians, emphasizing irresponsible behavior on the roads.

Although gender differences can be regarded as a protective factor supporting cognitive reserve concerning certain processes, such as visual–spatial abilities, as is the case for patients with Alzheimer’s disease [45] or patients with right brain injury [46], in the case of risk-taking driving it seems to persist in a negative sense, making men more vulnerable to accidents, even in old age. Indeed, men commit more errors and violations on the road than women [47,48]. Nori et al. [49] found that drivers, despite gender, with superior spatial orientation skills exhibited a higher frequency of aggressive driving violations than drivers with lower spatial skills. Women tend to exhibit lower visuo-spatial skills than men when assessing their spatial abilities. In contrast, men are more confident about their sense of direction and considered better than women in spatial navigation. Literature reports that men outperform women in distance estimation, following directions, reading maps, and mental rotations skills (e.g., [50]).

It is reasonable to assume that the same may be true for our particular sample. This higher competence leads older adults to display overconfidence regarding their driving skills and vehicle handling capabilities, creating an overconfidence bias [51]. This bias involves overestimating one’s actual performance, leading men to engage in riskier behaviors because they are confident that their abilities exceed reality. These feelings may also endure among older drivers. This framework also takes into account gender variations. Studies have shown that both novice and experienced male drivers tend to exhibit higher scores in risky and aggressive driving styles compared to women. Conversely, women display higher patient and cautious driving scores [52,53].

In addition, recent data suggest a change in the driving behaviors of young people, indicating that women are exhibiting more risky driving behaviors traditionally associated with men [54]. Moreover, the differences between male and female driving behaviors have diminished over time. This shift in trend can be attributed to increased traffic exposure among young women and a rise in risk-taking behaviors. Several studies have highlighted these changes in driving habits among young individuals [14,55,56,57,58]. It will be interesting to see if and how these shifts in women’s driving behaviors will impact the elderly population in the future.

However, until now, this change has not been found to affect individuals over 65, who maintain distinct driving styles and risk tendencies based on gender. It is important to note that, in our sample, the difference in risk tendency persists, even with equal car use. Men tend to take more risks and report more accidents compared to women, and experience in driving does not improve this tendency; instead, it increases their confidence in their ability to handle a vehicle in risky situations. Additionally, this risk tendency is evident in pedestrian behavior, with men showing lower awareness of their psychophysical state than women.

Silva et al. [59] found differences in strength between males and females in a study involving European countries; nevertheless, men and women have been found to report different perceptions of health and, consequently, quality of life. For instance, research has discovered that older women report a substantial 25% more complaints regarding their quality of life, ranging from poor to very poor, in comparison to older men [60]. These differences are also evident in the observation that older men are more vulnerable to environmental factors than women and that higher levels of physical fitness have a more significant impact on men’s quality of life and health assessment compared to women. This is in line with recent data reported by Piccardi et al. [61] showing that being active is essential, regardless of the type of motor activity or intensity of activity, although some forms of motor activity involving recreational and social conditions (i.e., dancing) are better than others [62].

Finally, the high estimation of one’s abilities, always present in males, could also result in an overestimation of the ability to perceive risky situations. However, various studies have demonstrated that hazard perception abilities, crucial for anticipating dangerous situations, decline with age, resulting in slower reaction times and an increased risk of severe accidents [63,64]. Based on these considerations, in fact, older men aged 70 and over have higher mortality rates than women of the same age [65].

It is crucial to provide specialized training for elderly drivers in addressing these issues. Research indicates that, despite the naturally declining cognitive and sensory abilities that come with age, the performance of elderly drivers can be improved through targeted training interventions. For example, studies have demonstrated that training focused on enhancing hazard perception can substantially decrease reaction times, reducing the risk of accidents. These training programs can assist elderly drivers in preserving their driving independence while minimizing associated risks (e.g., [66,67,68,69,70,71,72,73,74,75,76,77,78,79]).

It is important to note that, while training programs have shown promise, their implementation should be carefully tailored to meet the diverse needs of the elderly population. Tailored training that considers individual differences in cognitive and sensory decline could enhance the effectiveness of these programs. Furthermore, integrating regular assessments and updates into these training programs may help address the progressive nature of age-related decline.

In summary, data indicate that training for elderly drivers could be an effective strategy for improving road safety, reducing the severity of accidents, and supporting the safe mobility of older individuals. However, policies and training programs must consider these aspects to adequately address the increasing population of elderly drivers and the associated risks. Future research should focus on refining these training interventions and exploring additional strategies to ensure they are both accessible and practical for elderly drivers.

## 5. Limitations and Future Perspectives

This study shows some limitations worth mentioning. First, it involved only self-report measures of risk perception, which can suffer from the effect of social desirability. Second, no psychological variables, such as personality traits, cognitive styles, and cognitive, emotional, and motivational factors, have been included in the study. Future research should embrace a more holistic approach by incorporating objective measures, such as driving simulations, while also examining the combined effects of gender and psychological factors to better understand the individual differences in risky driving behavior. Third, we should have considered the impact of familiarity with the environment on risk perception. Research has shown that individuals are more likely to take risks in familiar environments [80]. Indeed, according to the spatial cognition perspective [81], being familiar with the environment enhances people’s sense of competence and safety [82,83], leading to a greater willingness to take risks [49].

From a practical standpoint, road safety campaigns should specifically focus on male drivers, as they are more prone to engaging in risky driving behaviors. By emphasizing responsible practices, such as adhering to speed limits, we can position these actions as socially desirable behaviors within this demographic (e.g., [84,85]). However, it is important to test the effectiveness of such messages directly through experimental studies. In general, repeated exposure to various messages, influencing the connection between driver perceptions and social desirability during and after initial driver training, could encourage less risky driving behavior.

## Figures and Tables

**Figure 1 geriatrics-09-00136-f001:**
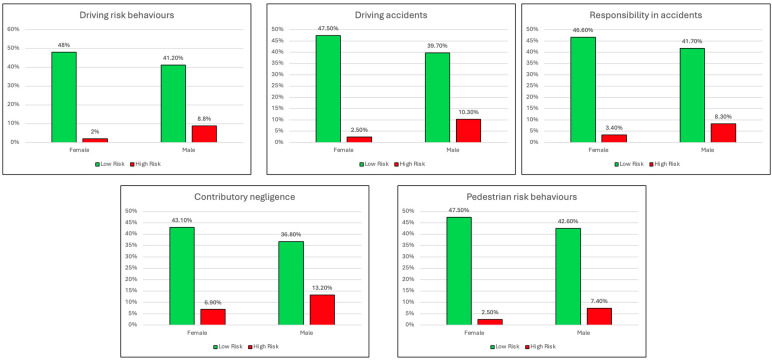
Summary of the odds ratio analysis.

**Table 1 geriatrics-09-00136-t001:** Descriptive statistics of the research sample by gender.

	Females (*N* = 99)				Males (*N* = 101)			
	Frequency (%)	M (SD)	Min- Max	z-Mean	Frequency (%)	M (SD)	Min- Max	z-Mean
*Age*		70.45	4.14	−0.03		70.64	4.61	0.01
*Education (years)*		11.97	4.05	0.12		11.04	3.81	−0.11
*Employment*								
Employed	1 (1%)				3 (3%)			
Retired	99 (99%)				97 (97%)			
*Marital status*								
Married	58 (57.40%)				80 (80.8%)			
Divorced	11 (10.9%)				8 (8.1%)			
Widowed	23 (22.8%)				7 (7.1%)			
Single	9 (8.9%)				4 (4.0%)			
*Years of holding driving license*		46.77	5.29	−0.08		48.61	7.11	0.17
*Weekly car use*		3.98	1.33	−0.09		4.21	1.19	0.10
Driving risk behaviors		1.75	1.60	−0.27		2.97	2.77	0.25
*Driving accidents*		0.96	1.23	−0.29		1.86	1.77	0.24
*Responsibility in accidents*		0.31	0.60	−0.21		0.61	0.88	0.17
*Contributory negligence*		0.16	0.42	−0.15		0.30	0.54	0.15
*Pedestrian risk behavior*		2.10	1.48	−0.24		3.00	2.25	0.22

**Table 2 geriatrics-09-00136-t002:** Inter-correlations among study variables.

	1.	2.	3.	4.	5.	6.	7.	8.	9.
1. Age	1	−0.07	0.41 **	−0.20 *	−0.11	−0.2	0.02	0.03	0.02
2. Education (years)	−0.03	1	0.18	−0.18	0.18	0.13	0.00	−0.4	0.29 **
3. Years of holding driving license	0.58 **	0.23 *	1	−0.17	−0.10	−0.08	−0.07	0.07	0.05
4. Weekly car use	−0.01	−0.07	−0.05	1	0.09	0.12	−0.04	0.22 *	−0.05
5. Driving risk behaviors	−0.02	0.15	0.17	−0.05	1	0.11	0.04	0.13	0.46 **
6. Driving accidents	−0.04	0.24 *	−0.02	0.04	0.03	1	0.63 **	0.46 **	0.11
7. Responsibility in accidents	0.11	0.011	0.10	0.08	0.13	0.67 **	1	0.25 *	0.14
8. Contributory negligence	0.04	−0.06	−0.03	0.03	0.12	0.38 **	0.33 **	1	0.06
9. Pedestrian risk behaviors	0.25 *	0.07	0.17	−0.15	0.50 **	0.21 **	0.15	−0.01	1

Note. Correlations for females (*N* = 101) are displayed above the diagonal; correlations for males (*N* = 99) are displayed below the diagonal. * *p* < 0.05 (two-tailed); ** *p* < 0.01 (two-tailed).

**Table 3 geriatrics-09-00136-t003:** Summary of the results of the odds ratio.

Variables	Groups	*N z*-Score < 1 *	*N z*-Score > 1 **	Odds Ratio	95% CI	*p*
Years of holding driving license	Female	97	5	2.83	0.97–8.27	0.048
	Male	89	13			
Weekly car use	Female	99	3	0.66	0.11–4.04	0.651
	Male	100	2			
Driving risk behaviors	Female	98	4	5.25	1.71–16.1	0.002
	Male	84	18			
Driving accidents	Female	97	5	5.03	1.82–13.9	<0.001
	Male	81	21			
Responsibility in accidents	Female	98	7	2.71	1.07–6.86	0.030
	Male	85	17			
Contributory negligence	Female	88	14	2.26	1.11–4.63	0.023
	Male	75	27			
Pedestrian risk behaviors	Female	97	5	3.34	1.17–9.58	0.019
	Male	87	15			

Note. *N* =200. * Number of participants with a score below 1 *z*-score positive (84.1% of the sample; Low Risk); ** Number of participants with a score above 1 *z*-score positive (15.9% of the sample; High Risk).

## Data Availability

Data are available on request by writing to the corresponding authors.

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
