# Peer review of "On the Road Safety: Gender Differences in Risk-Taking Driving Behaviors Among Seniors Aged 65 and Older"

_geriatrics, 2024, doi:10.3390/geriatrics9050136_

Round 1

Reviewer 1 Report

Comments and Suggestions for Authors

An important and interesting study. It provides comprehensive statistics and findings on the use of a number of useful research scales with older samples.  Its findings are of high relevance to both workers in geriatric research and in road safety. My only major concern is that the sample is relatively small, and the article would have been strengthened considerably if it included more respondents.  For example, larger numbers would have helped in the use of the study in comparable research. A larger sample would also have provided information for further development in this important area of road safety. For example, if the sample size had enabled the descriptive statistics to be reported for males and females and discussed separately in detail.  The correlation matrix clearly demonstrates an important gender linked increased risk.

Some additional information that would assist the reader includes-an extension of the explanation of the ageing rate--the relevant references are provided but it would clarify the issue to the reader if the terms "94.1% and 125.8%" (line 245) were spelt out in a more familiar way.

One minor spelling that needs attention-- "weakly" should be "weekly"

The other important issue that should be mentioned in the limitations of the research or as a need for further study is the under researched issue of the possible impairment of older persons' driving due to medication use. This presumably has an impact on both male and female driving. Drug labelling usually includes a warning about possibly impaired driving, and ideally the prescriber provides similar advice however I think that there is only limited research on their impact both on driving safety and on user caution compared with a large literature on illicit drug impairment. Some information on this issue would have made an important extension to the present study.

Comments on the Quality of English Language

The English is clear and well expressed.  I have noted some minor editing in the previous comment.

Author Response

We would like to extend our sincere thanks to the reviewer for their thoughtful and constructive feedback. We have taken all of the comments into account, and we believe that the revisions have substantially improved the manuscript. All corrections made to the manuscript are highlighted in yellow. Below is a detailed list of responses to your suggestions.

R: …My only major concern is that the sample is relatively small, and the article would have been strengthened considerably if it included more respondents.  For example, larger numbers would have helped in the use of the study in comparable research. A larger sample would also have provided information for further development in this important area of road safety…

A: We thank the Reviewer for this observation. We fully agree that a larger sample would have provided the opportunity for more in-depth analyses, particularly regarding gender differences. However, we would like to highlight that our study encountered  specific challenges in recruiting participants over 65 years old who met the established criteria (actively driving, regularly using a car, and without debilitating health conditions). While our survey was initially administered to 340 respondents, after data cleaning and gender balancing, we arrived at a final sample of 200 subjects (100 males and 100 females).

Furthermore, despite the sample size, our analyses conducted with G*Power demonstrates that the sample is still adequate for the correlations between variables and for the odds ratios, ensuring sufficient statistical power to detect significant effects with the expected effect size (r = 0.5 for the correlations, 25.9% of high-risk events for the odds ratios).

Additionally,  compared to our previous article, we replaced the Mann-Whitney U Test with Odds Ratio analysis in our gender comparison. This decision was motivated by the desire to provide a more direct and interpretable measure of the likelihood of high-risk behaviors between males and females, offering  more precise insights into relative risks associated with gender.

R: Some additional information that would assist the reader includes-an extension of the explanation of the aging rate-the relevant references are provided but it would clarify the issue to the reader if the terms "94.1% and 125.8%" (line 245) were spelt out in a more familiar way.

A:  Following the reviewer's recommendations, we have added more details to enhance the reader's understanding of this trend. Although, we did not omit the percentages that are reported in the quoted references.

R: The other important issue that should be mentioned in the limitations of the research or as a need for further study is the under researched issue of the possible impairment of older persons' driving due to medication use. This presumably has an impact on both male and female driving.

A: We are in complete agreement on this point. Particularly in our country, there is no proper attention to the consequences of drug use, while there is  adequate attention to the use of illegal drugs. In our research, we also selected a sample not subjected to any drug treatment. We have added that information in the sample description.

Thank you for your thorough review.

Reviewer 2 Report

Comments and Suggestions for Authors

The authors investigate risky driving behaviors among older adults, but do not sufficiently explain any gender differences as promised by the title.  Table 1 summarizes the data collected for the 200 participants, but it does not separate the data by gender.  Table 2 shows correlations between the variables, but again does not separate the correlations by gender.  In presenting test statistics, the authors show results of the Mann-Whitney U Test of differences between the variable scores for males versus females, but they do apply a simple test of mean differences between these groups.

Several major revisions are needed prior to re-review. Table 1 must be revised to show frequencies, means, and standard deviations by gender.  Table 1 must also include another column showing the z-statistics for differences in the means between genders. Table 2 must be separated by gender to show correlations and their significance levels by gender.

Sections 2.2.1 and 2.2.2 can be deleted without loss to the paper.  First, the two measures (DRPS and PBAPS) are not adequately described, and these two measures are never referred to anywhere else in the paper.  Second, the cluster analysis and bootstrapping applied to these measures blurs their contribution to the paper’s analysis. Third, the analysis of these measures does not compare the two genders, and no graphs are shown of the participants’ scores for these measures.

The U-statistic is not very informative, especially without seeing CDF graphs of the data rankings.  The authors should discard the analysis of U-statistics altogether, and instead report Odds Ratios of risky behavior by the two genders.  For each variable shown in Table 3, the authors can decide a threshold that separates the participants into two groups (less risky versus more risky).  The authors should also present graphs of the participants’ scores for each variable compared to the threshold lines.  That analysis will be much stronger and more useful than the current presentation.

The discussion of previous research both in the Introductions and Discussion is very thorough and helpful, but the data analysis of the paper must be revised.

Author Response

Reviewer 2

We would like to extend our sincere thanks to the reviewer for his thoughtful and constructive feedback. We have taken all comments into account and feel that the revisions have substantially improved the manuscript. In particular, we have introduced all the required static analyses. All corrections made to the manuscript are highlighted in yellow. Below is a detailed list of responses to suggestions.

R: Several major revisions are needed prior to re-review. Table 1 must be revised to show frequencies, means, and standard deviations by gender.  Table 1 must also include another column showing the z-statistics for differences in the means between genders. Table 2 must be separated by gender to show correlations and their significance levels by gender.

A: We have made all the required changes.

R: Sections 2.2.1 and 2.2.2 can be deleted without loss to the paper.  First, the two measures (DRPS and PBAPS) are not adequately described, and these two measures are never referred to anywhere else in the paper. Second, the cluster analysis and bootstrapping applied to these measures blurs their contribution to the paper’s analysis. Third, the analysis of these measures does not compare the two genders, and no graphs are shown of the participants’ scores for these measures.

A: We also agree with this criticism: the use of the two scales is unclear, mainly due to the lack of a detailed description in the manuscript. We revised the text to provide a more complete specification of the variables associated with these scales. Specifically, we have clarified that the DRPS scale measures ‘Driving Risk Behaviors,’ while the PBAPS scale assesses ‘Pedestrian Risk Behavior’. We believe that this enhanced description of the measurement tools now provides greater clarity.

R: The U-statistic is not very informative, especially without seeing CDF graphs of the data rankings.  The authors should discard the analysis of U-statistics altogether, and instead report Odds Ratios of risky behavior by the two genders.  For each variable shown in Table 3, the authors can decide a threshold that separates the participants into two groups (less risky versus more risky).

A: We have done the required analysis and actually feel that the results are now much clearer.

R: The authors should also present graphs of the participants’ scores for each variable compared to the threshold lines. That analysis will be much stronger and more useful than the current presentation.

A: We included a figure summarizing the Odd Ratio analysis (please, see Figure 1).

Thank you for your thorough review.

Round 2

Reviewer 1 Report

Comments and Suggestions for Authors

The points that could be considered for change are:

The sentence that is the last in the Abstract (lines 33-34) and is repeated in the second last sentence of section 5 (line 382) of  Limitations.  I would ask the authors to consider whether the phrase "view  responsible driving actions, such as observing speed limits , as socially desirable" is more appropriately expressed as  "less likely to view responsible driving actions, such as observing speed limits as desirable". "Desirable" could perhaps be more strongly expressed as "important".

Line 81 "intricate" could be better expressed as "complex" .

Comments on the Quality of English Language

Minor editing of English language required

Author Response

Changes have been highlighted in yellow colour

The sentence that is the last in the Abstract (lines 33-34) and is repeated in the second last sentence of section 5 (line 382) of  Limitations.  I would ask the authors to consider whether the phrase "view  responsible driving actions, such as observing speed limits , as socially desirable" is more appropriately expressed as  "less likely to view responsible driving actions, such as observing speed limits as desirable". "Desirable" could perhaps be more strongly expressed as "important".

Reply: We thank Reviewer’s for his/her suggestions we modified sentences and we asked for an editing of the English language.
Line 81 "intricate" could be better expressed as "complex" .

Reply: Done